# Catalytic Properties of Microporous Zeolite Catalysts in Synthesis of Isosorbide from Sorbitol by Dehydration

**Sangmin Jeong [1], Ki-Joon Jeon [1], Young-Kwon Park [2], Byung-Joo Kim [3] , Kyong-Hwan Chung [4] and Sang-Chul Jung [4],***

[1]  Department of Environmental Engineering, Inha University, 100 Inharo, Nam-gu, Incheon 22212, Korea; y1b2b5@naver.com (S.J.); Kjjeon@inha.ac.kr (K.-J.J.)
[2]  School of Environmental Engineering, University of Seoul, 163 Seoulsiripdaero, Dongdaemun-gu, Seoul 02504, Korea; catalica@uos.ac.kr
[3]  A Carbon Valley R&D Division, Korea Institute of Carbon Convergence Technology, 110-11 Banryong-ro, Jeonju 54853, Korea; kimbj2015@gmail.com
[4]  Department of Environmental Engineering, Sunchon National University, 255 Jungang-ro, Suncheon, Jeonnam 57922, Korea; likeu21@hanmail.net
*  Correspondence: jsc@sunchon.ac.kr; Tel.: +82-61-750-3814

**Abstract:** As bisphenol A has been found to cause hormonal disturbances, the natural biomaterial isosorbide is emerging as a substitute. In this study, a method for isosorbide synthesis from sorbitol was proposed by dehydration under high temperature and high pressure reaction. Microporous zeolites and Amberlyst 35 solid acids with various acid strengths and pore characteristics were applied as catalysts. In the synthesis of isosorbide from sorbitol, the acidity of the catalyst was the main factor. MOR and MFI zeolite catalysts with high acid strength and small pore size showed low conversion of sorbitol and low yield of isosorbide. On the other hand, the conversion of sorbitol was high in BEA zeolite with moderate acid strength. Amberlyst 35 solid acid catalysts showed a relatively high conversion of sorbitol, but low yield of isosorbide. The Amberlyst 35 solid acid catalyst without micropores did not show any inhibitory effects on the production of by-products. However, in the BEA zeolite catalyst, which has a relatively large pore structure compared with the MOR and MFI zeolites, the formation of by-products was suppressed in the pores, thereby improving the yield of isosorbide.

**Keywords:** isosorbide; solid acid catalyst; sorbitol; dehydration; bisphenol A

## 1. Introduction

Bisphenol A (4,4′-isopropylidenediphenol, BPA) is a monomer synthesized by the condensation of acetone with two molecules of phenol. BPA is a typical female hormone disruptor and is a raw material of epoxy resin or polycarbonate. BPA has also been used as a coating material to prevent cans from corrosion, or as a plastic additive to increase heat resistance and durability. Polycarbonate has high heat resistance and transparency and is widely used for household goods such as water bottles, baby bottles, food storage containers, and industrial products such as CDs. In addition, BPA is used in a number of products used in daily life such as disposable paper cups and thermal paper receipts. However, several studies have raised the risks of BPA including endocrine disruption, metabolic disorders, hypertension, and premature maturation [1,2]. When food is stored in packaging using BPA, a report suggested that BPA may elute and affect people through food, limiting the use of BPA [3].

In particular, infants were considered to be sensitive to BPA exposure, and their use was completely prohibited [4–9].

Due to the potential risks of BPA, various attempts have been made to replace BPA as regulations progress. The first attempts were made to utilize the same bisphenol-based materials, bisphenol E (BPE), bisphenol S (BPS), and bisphenol F (BPF) [10]. The study on amphibians has shown that BPA inhibits the activity of γ-secretase in early amphibian embryos. However, this phenomenon did not occur when treated with the same concentration of BPE and BPF [11]. Although BPA requires two methyl groups to inhibit γ-secretase activity, it was concluded that BPE and BPF did not have teratogenic effects such as BPA because there was only one methyl group in each. BPS and BPF have almost the same physical properties as BPA, and have been used in many products such as can coatings, thermal paper receipts, and polycarbonates [12].

Indeed, the dangers of BPA have been emphasized and as an alternative, BPA-free products have begun to be used. However, BPA, BPF, and BPS have similar chemical structures to BPA, and these alternatives have also been shown to cause endocrine disruption [13]. Papers comparing BPF and BPA reported the same anti-androgen activity in proportion to the concentration of both substances in vitro [14–17]. The same risks as BPA were seen in BPS as well as in BPF. Recently published studies have reported a 180–240% increase in neuronal development near the hypothalamus of zebra fish, even at very low levels (0.0068 μM) of BPA and BPS, suggesting a link between BPA and BPS.

BPA, phthalates, and nonylphenols, which are the major environmental hormone obstacles, are the main raw materials of plastics and detergents and have a high exposure in daily life [18]. Recognition of the dangers of environmental hormonal barriers is driving global regulations on these substances and replacing existing products with products that use less dangerous ingredients. However, since there are no clear alternatives right now, these environmental hormonal obstacles are still being used. Research continues to raise concerns about alternatives, as some alternatives have been shown to present similar risks to existing environmental hormone barriers [19].

As BPA substitutes using synthetic compounds still show endocrine disrupting effects, efforts have been made to find alternatives in natural products. Sorbitol is a major source of biomass as a useful chemical [20–24] and is considered one of the 10 most important sources of biomass [25]. Generally, sorbitol is produced through hydrogenation of glucose derived from starch. In the last decade, much effort has been made to produce sorbitol from cellulose through hydrolysis hydrogenation. Much research has expanded the potential of sorbitol as a bio-based feedstock and is being produced in high yield from cellulose [26–36].

Isosorbide is a 100% natural biomaterial made from corn. Isosorbide (1,4:3,6-dianhydro-D-sorbitol) is produced through the dehydration of the double molecule of sorbitol. Starch is extracted from corn and is made by glucose, sorbitol, etc. Isosorbide is produced by enzymatic glycosylation of starch from corn, followed by hydrogenation, followed by dehydration. Isosorbide is widely used in a variety of industries such as the pharmaceutical industry due to its high stability, two symmetric OH groups, and its unique properties [37,38]. Another important application of isosorbide is that it is used as a plastic monomer [39,40]. Poly(ethylene-co-isosorbide) terephthalate is a bio-based alternative to polyethylene terephthalate (PET) and has a higher glass transition temperature than PET [41]. Isosorbide is also attracting attention as a highly functional material that can replace BPA in polycarbonate and epoxy resin production [42].

Plastics made of isosorbide as the raw material have distinguished advantages such as superior transparency and surface hardness as well as eco-friendly properties such as biodegradability and nontoxicity, compared to plastics made of petrochemicals. It is expected to be widely used for exterior materials of electronic devices such as mobile devices and TVs, liquid crystal films of smart phones, automobile dashboards, food containers, and eco-friendly building materials. Isosorbide, which can be synthesized from plants such as corn, is attracting attention as an environmentally friendly alternative due to its high biodegradability and excellent heat resistance [43].

Zeolites have well-defined structures that are microporous crystalline solids. Generally, they are composed of silicon, aluminum, and oxygen in their framework and cations. They have been applied in many fields of catalysis, generating intense interest in these materials in industrial and academic laboratories. They present appreciable acid activity with shape-selective features as catalysts [44,45]. The FAU zeolite consists of 12-oxygen rings, with pores with a of diameter 7.4 Å in a tetrahedral configuration, with large pores called supercages at the intersection. The MFI zeolite has an unusual structure in which straight pores with 5.4 × 5.6 Å cross sections and zig-zag pores with 5.1 × 5.5 Å intersect each other. The pores of the MWW zeolite consist of two independent 10-oxygen rings. The two-dimensional 10-oxygen ring is sinusoidal and the other is composed of a 10-oxygen ring and 12-oxygen ring. The pores of the BEA zeolite consisted of 12-oxygen rings, and the pore size parallel to the [001] plane was small, but the pore size of the [100] plane was large. The pores were curved like sinusoids. The MOR zeolite has a straight main hole consisting of 12-oxygen rings and pores consisting of 8-oxygen rings there between.

In this study, we propose a method for isosorbide synthesis that is drawing attention as an alternative to BPA. Sorbitol is used as a raw material to investigate the reaction characteristics of isosorbide synthesis and to find the optimum catalyst. Various zeolites were introduced as a solid acid catalyst. The dehydration reaction of sorbitol, according to the physicochemical properties of the catalyst, and the synthesis process of isosorbide are discussed.

## 2. Results and Discussion

### 2.1. Characteristics of the Catalysts

The X-ray diffraction pattern of the zeolite catalyst used in the reaction is shown in Figure 1. The characteristic diffraction peaks of the synthesized zeolites agreed well with the position and size ratio of the characteristic diffraction peaks presented in the literature [46]. SEM images of the zeolites are shown in Figure 2. The particles of the BEA, FAU, MFI zeolite were small regular cubes. The crystal size of the MOR zeolite was rod-shaped, about 1.5 μm. The surface of the MWW zeolite was lumpy in shape, very tangled. The particle size of BEA zeolite was very small, 0.2 μm or less. The FAU and MFI zeolites were uniform in size, approximately 0.5 μm. Figure 3 shows the nitrogen adsorption isotherms of the zeolite catalysts. Zeolites used in experiments such as the MFI zeolite are zeolites with microporous development and have a typical Langmuir adsorption isotherm pattern due to the large number of micropores. On the other hand, the relatively large pore size and the wide surface area of the BEA zeolite increased the adsorption amount at a relative pressure of 0.2~0.7. The reason for the large adsorption amount in the BEA and FAU zeolites is that the pore size of the zeolite is large and there is a lot of empty space such as a supercage in the pore.

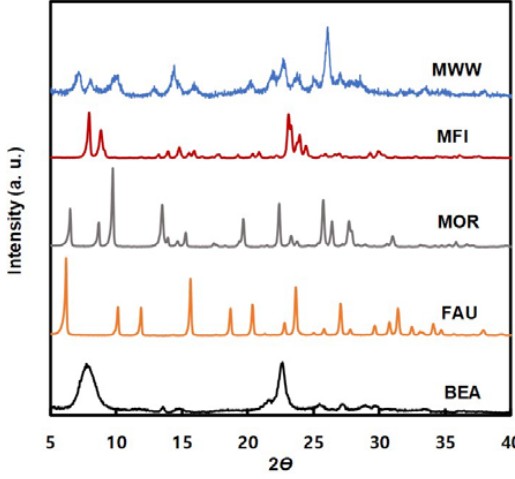

**Figure 1.** XRD patterns of the MWW, BEA, MOR, FAU, and MFI zeolites.

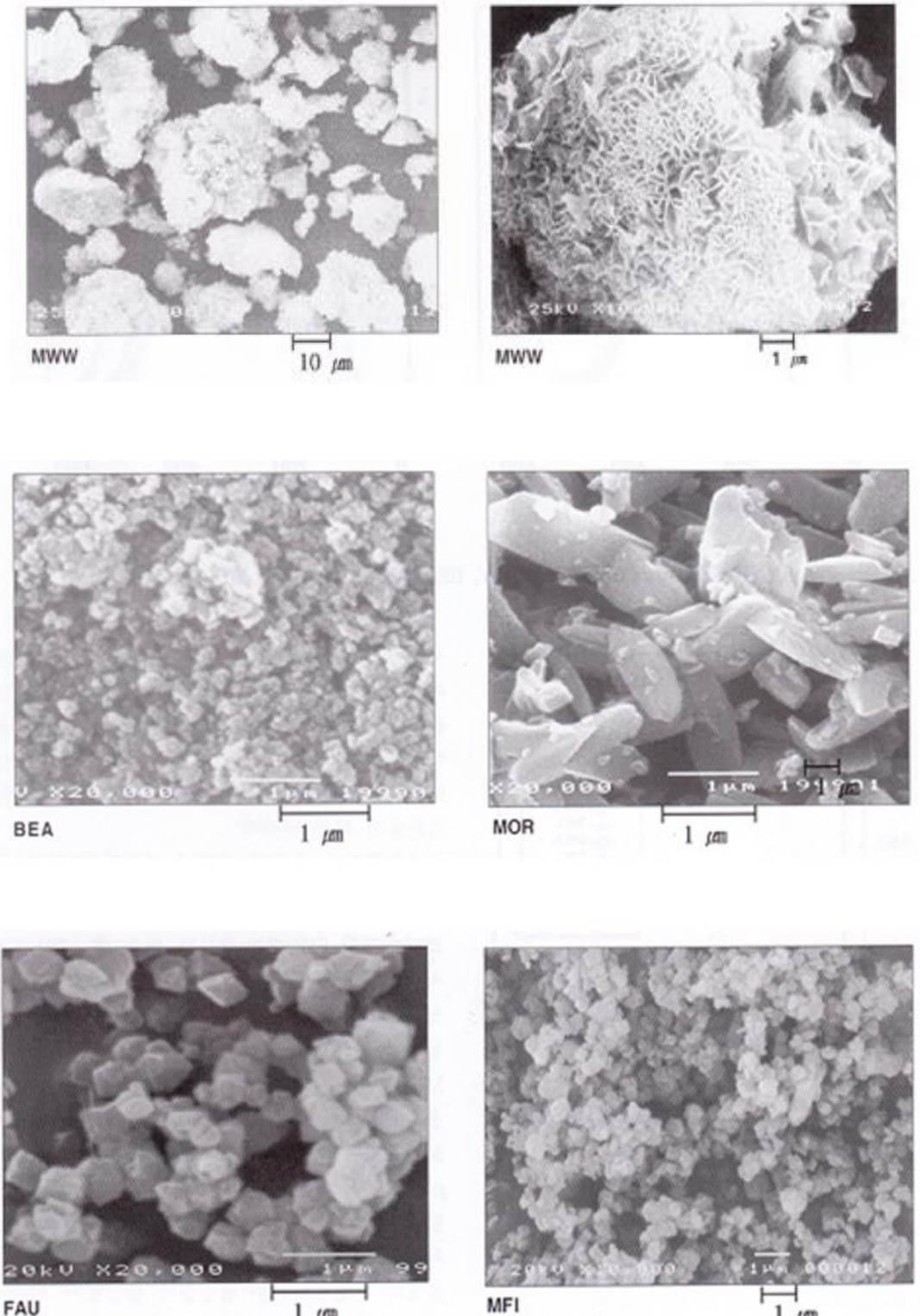

**Figure 2.** SEM images of the MWW, BEA, MOR, FAU, and MFI zeolites.

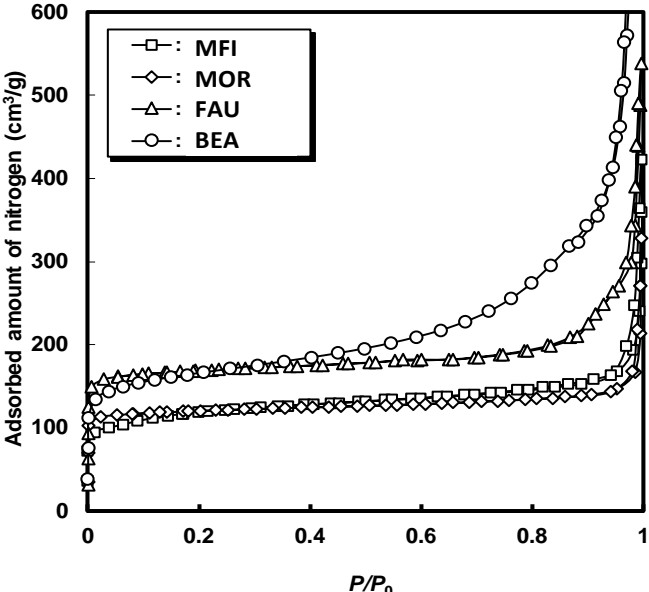

**Figure 3.** NH$_3$-temperature programmed desorption (TPD) profiles of the various zeolites.

The BET surface area and pore structure parameters obtained from nitrogen adsorption isotherms are summarized in Table 1. Unlike other zeolites such as MOR, the MFI zeolite has a pore size of 10-membered oxygen rings, and thus the pore size is smaller than that of other zeolites. MOR zeolites have a straight pore structure, whereas MFI zeolites are bent in a zigzag shape. The BET surface area was larger in the order of BEA > FAU > MOR > MFI zeolite.

**Table 1.** Physical properties of the zeolite catalysts used in this work.

| Zeolite | Si/Al Molar Ratio (-) | Pore Diameter (Å) | BET Surface Area (m$^2$/g) | Micropore Volume [1] (cm$^3$/g) |
|---------|-----------------------|-------------------|-----------------------------|----------------------------------|
| MWW | 13 | 5.6 × 5.6 | 420 | 0.15 |
| BEA | 13 | 7.6 × 6.4, 5.5 × 5.5 | 690 | 0.19 |
| FAU | 5 | 7.4 × 7.4 | 700 | 0.24 |
| MOR | 10 | 6.5 × 7.0, 2.6 × 5.7 | 410 | 0.13 |
| MFI | 50 | 5.3 × 5.6, 5.1 × 5.5 | 260 | 0.12 |

[1] determined from the t-plot method.

The NH$_3$-TPD results are shown in Figure 4 to compare the acidity of zeolites from the ammonia desorption curve. Only one bud appeared in the TPD curve of the FAU zeolite shown on the left side of the figure. On the other hand, the TPD curves of other zeolites can be divided into two desorption buds around 200 °C and 300–500 °C. It is known that buds appearing at low temperatures are due to the desorption of physisorbed ammonia and buds appearing at high temperatures due to the desorption of ammonia adsorbed at strong acid sites [47]. The lower temperature and smaller size of the second desorption bud of the MWW and BEA zeolite means weak and low acidity. Acid strength can be estimated by the maximum peak temperature ($T_{max}$) of the desorption peak associated with the activation energy for ammonia desorption [48]. The acid strength of the MWW zeolite was similar to that of the MFI zeolite. The order of acid strength was FAU ≒ BEA < MWW ≒ MFI < MOR. MOR zeolites had the highest acidity compared to other zeolites, with the highest peak temperature of the second desorption bud at 480 °C. The MFI zeolite had the largest number of scattering points as the second detachable bud was the largest.

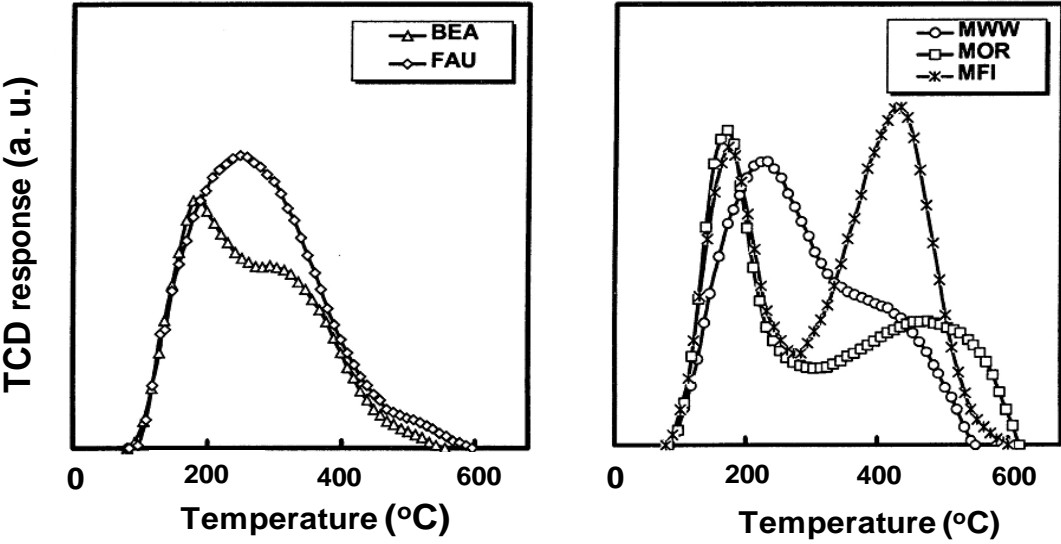

**Figure 4.** NH$_3$-TPD profiles from the MWW, BEA, MOR, FAU and MFI zeolites.

Figure 5 shows the adsorption behavior of *o*-xylene on the MWW, BEA, MOR, FAU, and MFI zeolites. The behavior shows the adsorption of *o*-xylene inside the pores of the zeolite. *o*-xylene was rapidly adsorbed on BEA and FAU with a large pore space. The *o*-xylene adsorption levels at MOR and MWW were lower than for the BEA and FAU zeolite catalysts. Adsorption of *o*-xylene on the MFI zeolite with narrow pore entrance and curved pore structure was slower than that of other zeolites. This suggests that steric hindrance due to various pore structures of zeolites can affect the reaction.

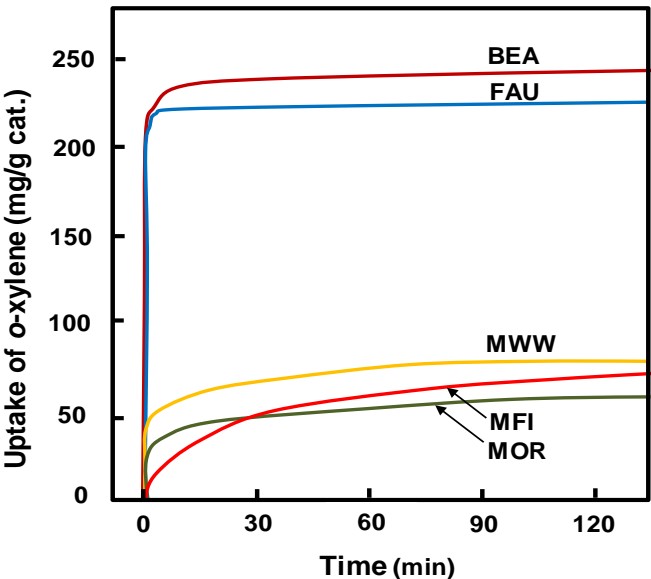

**Figure 5.** Adsorption of *o*-xylene on the MWW, BEA, MOR, FAU, and MFI zeolites.

### 2.2. Reaction Characteristics of Isosorbide Synthesis on the Catalysts

As shown in Scheme 1, sorbitol is dehydrated to either 1,4-sorbitan, 1,5-sorbitan, or 2,5-sorbitan. 1,4-sorbitan reacts further to isosorbide, while 1,5- and 2,5-sorbitan did not exhibit in the high-performance liquid chromatography (HPLC) chromatogram. Analytical peaks of the products analyzed by HPLC are shown in Figure 6. Detection peaks for the raw and reactant sorbitol, the intermediate product 1,4-sorbitan, and the desired product isosorbide were observed. Figure 7

shows the conversion of sorbitol and the yield of isosorbide according to the reaction temperature in the BEA zeolite catalyst. At the reaction temperature in the range of 160 °C to 220 °C, the internal pressure of the autoclave ranged from 7 bar to 10 bar. The conversion of sorbitol and the yield of isosorbide were the highest at the reaction temperature of 200 °C and decreased with increasing reaction temperature. If the reaction temperature is higher than 200 °C, it appears that the catalyst deactivation occurs by carbon deposition.

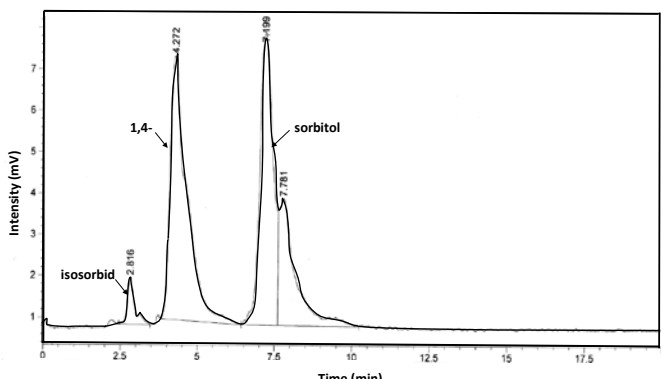

**Scheme 1.** Reaction scheme of isosorbide synthesis from sorbitol dehydration.

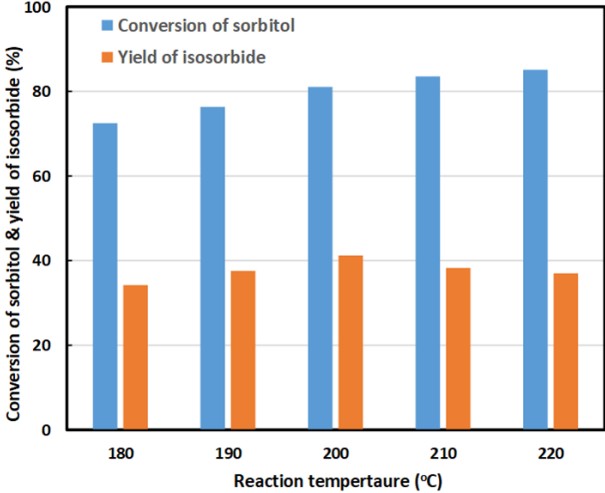

**Figure 6.** High-performance liquid chromatography (HPLC) chromatogram of products.

**Figure 7.** Conversion of sorbitol and yield of isosorbide by dehydration reaction over the BEA zeolite catalyst.

Figure 8 shows the change of sorbitol conversion and isosorbide yield in the BEA and FAU zeolites and Amberlyst 35 solid acid catalyst. The conversion of sorbitol and isosorbide yield in the MFI zeolite with high acid strength, but small pore size was lower than that of the BEA zeolite or Amberlyst 35. In comparison, the BEA zeolite showed the highest sorbitol conversion and isosorbide yield among the applied catalysts. The conversion of sorbitol with a relatively large molecular size was higher in the BEA zeolite with a larger pore size than the MFI zeolite with a smaller pore size and higher reactivity at medium acid strength than strong acid strength. Amberlyst 35 catalysts had relatively high conversions, but slightly lower yields of isosorbide. It is suggested that this was caused by not showing the effect of inhibiting the formation of by-products by the pores in the conversion reaction. Table 2 summarizes the sorbitol conversion and isosorbide yields investigated by applying various acid catalysts. As can be seen from this result, the BEA zeolite showed the best reactivity in this synthesis reaction. If the acid strength of the catalyst was too strong, the conversion of sorbitol and the yield of isosorbide were very low. The conversion of sorbitol was high in the catalyst with a medium acid strength. MOR and MFI zeolite catalysts with high acid strength and small pore size showed a very low conversion of sorbitol and yield of isosorbide. The conversion of sorbitol was the highest in the BEA zeolite catalyst and the yield of isosorbide was higher than that of other catalysts. 1,4-sorbitan was mainly obtained as a by-product. It was determined that the yield was increased because the production of by-products during the reaction was suppressed by the steric effect of the pore structure of the BEA zeolite.

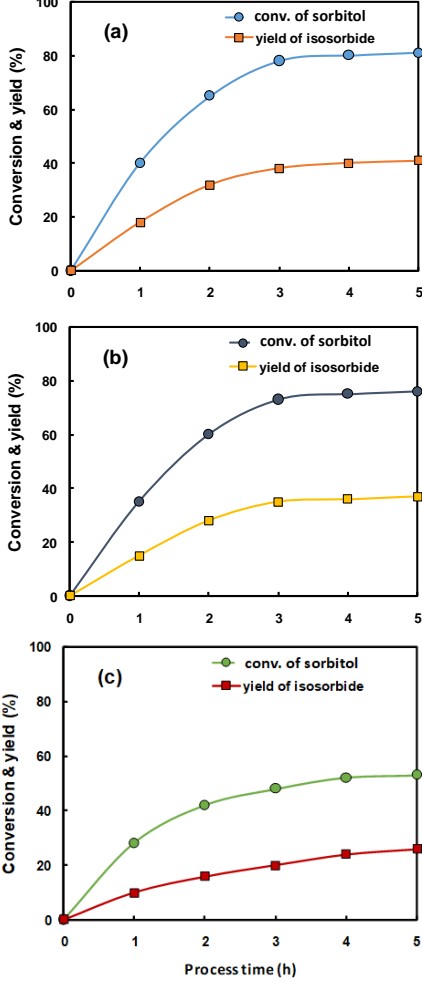

**Figure 8.** Conversion of sorbitol and isosorbide yield as a function of process time on the (**a**) BEA zeolite, (**b**) Amberlyst 35, and (**c**) FAU zeolite.

**Table 2.** Synthesis of isosorbide by dehydration of sorbitol on various catalysts after 5 h of process time at 200 °C.

| Catalyst | Conversion (%) | Yield (%) |
|---|---|---|
| BEA | 81.1 | 41.2 |
| MFI | 39.1 | 18.5 |
| FAU | 53.0 | 26.1 |
| SBA-15 | 10.1 | 5.2 |
| MWW | 15.7 | 11.2 |
| Amberlyst 35 | 76.1 | 36.8 |

## 3. Experimental

### 3.1. Materials and Catalysts

D-sorbitol (99%, Sigma-Aldrich, St. Louis, MO, USA) was used as a reactant, and mixed with a catalyst for the dehydration reaction in an autoclave reactor. The catalysts were zeolite acid catalysts with micropores and an Amberlyst 35 (Sigma-Aldrich, St. Louis, MO, USA) solid acid catalyst. The zeolites were MOR, MFI, FAU, BEA, and MWW zeolite catalysts with different pore sizes and different acidity. MOR zeolites (Si/Al = 10)) with a Si/Al molar ratio of 10 was purchased from Tosoh. To convert to the H-type MOR zeolite, $Na^+$ in MOR zeolite was ion-exchanged with a solution of 0.5 M ammonium nitrate (>99 wt%, Duksan, Seoul, Korea) at 60 °C and calcined at 550 °C for 6 h to form H-type MOR zeolite.

The MFI zeolite was synthesized from the liquor containing a mixture of colloidal silica (Ludox, 40 wt% $SiO_2$, Sigma-Aldrich, St. Louis, MO, USA), aluminum hydroxide (64 wt%, Sigma-Aldrich, St. Louis, MO, USA), potassium hydroxide (80 wt%, Duksan, Seoul, Korea), and secondary distilled water. After aging for 12 hours, the mixture was heated at 190 °C for two days in a high pressure reactor. The Si/Al molar ratio of the MFI zeolite synthesized by the concentration of the synthetic mother liquor was 25. The MWW (Si/Al = 13) zeolite was prepared according to the methodology reported elsewhere [49]. The ion exchange process for converting the H-type MFI zeolite was applied in the same manner as in the preparation of the MOR zeolite. BEA and FAU zeolites were prepared by ion-exchanging Na-BEA (Si/Al = 13, Zeolyst Co., Kansas City, USA) and Na-FAU (Si/Al = 5, Zeolyst Co., Kansas City, USA) in the same manner as above. BEA and FAU zeolites were also washed, dried, and calcined. The zeolite used in the experiment was named MOR, MFI, FAU, BEA, and MWW according to the International Zeolite Association (IZA) code name, and the Si/Al molar ratio was written in parentheses after the name. SBA-15 was synthesized according to the methods presented in the literature [50].

### 3.2. Preparation of Isosorbide from Sorbitol

Isosorbide was synthesized by the dehydration reaction at high temperature and high pressure. As a reactant, 0.5 g of D-sorbitol (99%, Sigma-Aldrich, St. Louis, MO, USA) was dissolved in 20 mL of distilled water, followed by reaction with 0.1 g of a catalyst. The reactor used a batch stainless autoclave capable of temperature maintenance and magnetic stirring. The reactant and the catalyst were added to the reactor and stirred at 500 rpm. The dehydration reaction was carried out in the temperature range of 180 °C to 220 °C. The composition of the product produced by the reaction was analyzed for composition using high-performance liquid chromatography (HPLC; Shimadzu, LC-20A, Kyoto, Japan) equipped with a refractive index detector and a REZEX RCM-monosaccharide column.

### 3.3. Characterization of the Catalysts and Products

The X-ray diffraction (XRD) pattern of the zeolite was irradiated with an X-ray diffractometer (D/MAX Ultima III, Rigaku, Tokyo, Japan). CuKα X-rays passed through the Ni-filter at 40 kV and 40 mA conditions were injected at a rate of 2°/min over a range of 5–50°. The particle size and

shape of the zeolite was observed by scanning electron microscope (SEM; Hitachi, S-4700, Tokyo, Japan). The Si/Al molar ratio was calculated by measuring the silicon and aluminum content with energy-dispersive X-ray spectroscopy (EDX; Horiba, EX-250, Tokyo, Japan) mounted on the SEM. Nitrogen adsorption isotherms were measured by a nano-porosity analyzer (Nano Porosity Analyzer, Mirae SI Co., nanoPOROSITY-XQ, Gwangju, Korea). After exhausting at 150 °C for 2 h, nitrogen was adsorbed at −196 °C. Surface area was calculated using the Brunauer–Emmett–Teller (BET) equation.

In order to investigate the acidity of the zeolite catalyst, the ammonia elevated temperature desorption ($NH_3$-TPD) curve was drawn using a temperature desorption test apparatus (Bel Japan, BELCAT, Osaka, Japan). The catalyst was evacuated at 550 °C for 1 h in a helium stream and then cooled to 150 °C. Ammonia gas was sent in pulses to the catalyst and adsorbed until saturated. In order to remove the physically adsorbed ammonia, it was exhausted while flowing a helium stream for 1 h, and the ammonia desorbed while analyzing the temperature. The temperature was raised to 600 °C at a rate of 10 °C/min. The amount of the acid point of the zeolite catalyst was calculated by deconvolution of the measured TPD curve to obtain the amount of weak and strong acid points.

## 4. Conclusions

Sorbitol was used as a raw material to prepare isosorbide by high temperature and high pressure reaction. Sorbitol conversion was higher for the BEA zeolites and Amberlyst 35 solid acid catalysts with moderate acid strengths than for the acidic MOR or MFI zeolites. The yield of isosorbide was not high in the MOR or MFI zeolite catalysts with a small pore size, but relatively high in the BEA zeolite catalyst. It was shown that the relatively large molecular size of sorbitol was enhanced by the inhibitory effect of the byproduct formation in the BEA zeolite catalysts with pores of similar size to the reactant and product molecular sizes than the zeolite catalysts with small pore sizes. The Amberlyst 35 solid acid had a relatively high conversion of sorbitol, but not a high yield of isosorbide. This was caused by not showing the effect of inhibiting the formation of by-products by pores in the conversion reaction. In contrast, the BEA zeolite was found to increase the yield because it suppresses the formation of by-products by the pore structure of the catalyst.

**Author Contributions:** S.J., K.-H.C., and S.-C.J. designed the experiments. S.J. and K.-H.C. performed the experiments and wrote the original drafted paper. K.-J.J. interpreted the data. Y.-K.P. and B.-J.K. contributed to the analysis. S.-C.J. supervised the experiments and paper. All authors discussed the results and contributed to the manuscript. All authors have read and agreed to the published version of the manuscript.

**Funding:** We do not have Funding.

**Conflicts of Interest:** The authors declare no conflict of interest.

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
