# Peer review of "Catalytic Properties of Microporous Zeolite Catalysts in Synthesis of Isosorbide from Sorbitol by Dehydration"

_catalysts, doi:10.3390/catal10020148_

Round 1
Reviewer 1 Report
The authors reported the conversion of sorbitol to isosorbide by using several kind of Zeolites.
Remarks
it is not clear the reson of referiment to bisphenol A. in the text
Bisphenol A is a monomer that remain entrapped in polycarbonate after the synthesis with phosgene to obtain polycarbonate. Polycarbonate is used as transparent material for various optical use as sunglasses and helmets, parts of aircraft. If we consider Bisphenol A as cover of metals, is however not clear if the derivate of sorbitlo will employ to produce composite.
Title: Title reported in the some sentenze the word catalyst is suggested to uniform the text.
Line 17-19. Sentence not clear from " BPA A"
Line 27: Relatively Large pore define reduce surface area.
Line 33. sentence non clear: it is a typical BPA ormon distruptor typically female hormon
Line 34. polycarbonate used for food packaging and epoxy resin
this Sentence not cleaar
Line 37. add. "gases" to water bottles, food storage container and milk to bady bottles.
Line 40. Endocirne disorder premature is indicate to introduce reference
Line 41-42. use of BPA that affect people
Line 42. Please mention cardiac effect of BPA
Line 53-56 anti androgen activityes
Line 63. Sentence not clear
Line 70. sorbitole: more used biomass
Line 86. Plastic materials transparency
Line 110. Sentence "large pore size large surface area"
Line 117. Sequence not clear pore size of 10 oxygen bond structures.
Line 138 decomposition of xilene. it is not clear how the confinement of o- Xilene define decomposition in zeolites.
Line 156 Figure HPLC. low quality of picture. is required the reference in the picure
Line 114 the quality resolution required to be improved.
Reference
This reference reported the conversion of sorbitolo to isosorbide.
https://pubs.rsc.org/en/content/getauthorversionpdf/C5GC00319A
some contents seem to be deducted from the text
no respected order of presentation in the manuscrip. This define a difficult to read the text.
Author Response
Dear Editor and Reviewers
Thank you for allowing the opportunity to respond to reviewers’ comments regarding the above-referenced manuscript. The manuscript was revised according to the reviewers’ comments. Changes made in response to the comments are described below.
==================================================================
Reviewer 1
it is not clear the reson of referiment to bisphenol A. in the text Bisphenol A is a monomer that remain entrapped in polycarbonate after the synthesis with phosgene to obtain polycarbonate. Polycarbonate is used as transparent material for various optical use as sunglasses and helmets, parts of aircraft. If we consider Bisphenol A as cover of metals, is however not clear if the derivate of sorbitlo will employ to produce composite.
Title: Title reported in the some sentenze the word catalyst is suggested to uniform the text.
Line 17-19. Sentence not clear from " BPA A"
Line 27: Relatively Large pore define reduce surface area.
Line 33. sentence non clear: it is a typical BPA ormon distruptor typically female hormon
Line 34. polycarbonate used for food packaging and epoxy resin
this Sentence not cleaar
Line 37. add. "gases" to water bottles, food storage container and milk to bady bottles.
Line 40. Endocirne disorder premature is indicate to introduce reference
Line 41-42. use of BPA that affect people
Line 42. Please mention cardiac effect of BPA
Line 53-56 anti androgen activityes
Line 63. Sentence not clear
Line 70. sorbitole: more used biomass
Line 86. Plastic materials transparency
Line 110. Sentence "large pore size large surface area"
Line 117. Sequence not clear pore size of 10 oxygen bond structures.
Line 138 decomposition of xilene. it is not clear how the confinement of o- Xilene define decomposition in zeolites.
Line 156 Figure HPLC. low quality of picture. is required the reference in the picure
Line 114 the quality resolution required to be improved.
Reference
This reference reported the conversion of sorbitolo to isosorbide.
https://pubs.rsc.org/en/content/getauthorversionpdf/C5GC00319A
some contents seem to be deducted from the text
no respected order of presentation in the manuscrip. This define a difficult to read the text.
[Response] As the authors mentioned in the introduction to this paper, plastics made of isosoride as raw materials have similar performance to polycarbonates. This means that transparency and robustness are similar to those of polycarbonate and are environmentally friendly. Isosorbide can synthesize sorbitol as a raw material. In this paper, we propose a method for the production of isosorbide that replaces bisphenol A, which is used to prepare polyates.
The authors corrected all of the typographical and typographical errors indicated by the reviewer. The revised contents were all reflected in the revised papers submitted. Figure 6 was redrawn with high quality in the revise manuscript. The authors appreciate the careful comments of the reviewer.
Reviewer 2 Report
The paper is very good written and is very interesting. As Bisphenol A has been found to cause hormonal disturbances, natural biomaterial isosorbide is emerging as a substitute. It was proposed the new method by dehydration under high temperature of sorbitol.
But it is nessesary to improve english and also some recomendations.
1.It is necessary to study the adsorption processes in more detail and to study the information about the activation energy of molecules.
2. Reason the choice of different zeolites
3. Specify in detail how the silicate module on zeolites was defined.
Author Response
Dear Editor and Reviewers
Thank you for allowing the opportunity to respond to reviewers’ comments regarding the above-referenced manuscript. The manuscript was revised according to the reviewers’ comments. Changes made in response to the comments are described below.
==================================================================
Reviewer 2
The paper is very good written and is very interesting. As Bisphenol A has been found to cause hormonal disturbances, natural biomaterial isosorbide is emerging as a substitute. It was proposed the new method by dehydration under high temperature of sorbitol. But it is nessesary to improve english and also some recomendations.
1.It is necessary to study the adsorption processes in more detail and to study the information about the activation energy of molecules.
[Response] The authors described the adsorption isotherm of nitrogen on zeolites in more detail, as reviewers pointed out. The surface area and pore size due to adsorption characteristics are also described. The investigation of the adsorption process of o-xylene on zeolites is intended to infer the sorbitol reaction activity from the pore adsorption characteristics of o-xylene with similar molecular size to the reactants. From this, the reaction characteristics of the zeolites with different pore sizes were investigated. The authors appreciate the reviewers' valuable comments.
Reason the choice of different zeolites Specify in detail how the silicate module on zeolites was defined.
[Response] The authors described the characteristics of zeolites in the text according to the reviewer's comments. The reason why the zeolites with different acid strengths and different pore sizes were applied to this experiment was to identify the main factors affecting the reaction in the synthesis of isosorbide from sorbitol. The following is a supplement to the revised paper.
“Zeolites have well-defined structures with are microporous crystalline solids. Generally, they are composed of silicon, aluminum, and oxygen in their framework and cations. They have been applying in many fields of catalysis, generating intense interest in these materials in industrial and academic laboratories. They present appreciable acid activity with shape-selective features as catalysts [44,45]. FAU zeolite consists of 12-oxygen rings, with pores of diameter 7.4 Å in tetrahedral configuration, with large pores called supercage at the intersection. MFI zeolite has an unusual structure in which straight pores with 5.4 x 5.6 Å cross sections and zig-zag pores with 5.1 x 5.5 Å intersect each other. The pores of the MWW zeolite consist of two independent 10-oxygen rings. The two-dimensional 10-oxygen ring is sinusoidal and the other is composed of 10-oxygen ring and 12-oxygen ring. The pores of the BEA zeolite consisted of 12-oxygen rings, and the pore size parallel to the [001] plane is small but the pore size of the [100] plane is large. The pores are curved like sinusoids. MOR zeolite has a straight main hole consisting of 12-oxygen rings and pores consisting of 8-oxygen rings there between.”
Reviewer 3 Report
Catalysts 693016
Title: “Catalytic properties of microporous zeolite catalysts in synthesis of isosorbide from sorbitol by dehydration”
In this paper, the authors show a method for the synthesis of isosorbide from sorbitol dehydration at high temperature and pressure. Microporous zelolites and Amberlyst 35 were applied as catalysts”
The chemistry of this paper is sufficiently interesting, but is necessary one major revision of all manuscript.
Comment:
Abstract
Define BEA zeolite catalyst
Revise all manuscript
Results and Discussion
Define BEA, FAU, MFI MOR and MWW zeolites. Difference between these zeolites. Comment the values of pore diameter (Å) with major detail. Figure 4
Because the separation in two figures. BEA, FAU and MWW, MOR, MFI
Revise the presentation of the Figure 6 It is necessary that the authors comment with major detail Scheme 1. The authors comment the uses of “high acid”, which is the acid used.
Other comment:
Page 9, Line 189
CHANGE 0.5 N FOR 0.5 M
Page 9, Line 193
CHANGE SiO2 FOR SiO2
Page 9, Line 195
“After aging for hours” How many hours?
Author Response
Dear Editor and Reviewers
Thank you for allowing the opportunity to respond to reviewers’ comments regarding the above-referenced manuscript. The manuscript was revised according to the reviewers’ comments. Changes made in response to the comments are described below.
==================================================================
Reviewer 3
In this paper, the authors show a method for the synthesis of isosorbide from sorbitol dehydration at high temperature and pressure. Microporous zelolites and Amberlyst 35 were applied as catalysts”
The chemistry of this paper is sufficiently interesting, but is necessary one major revision of all manuscript.
Comment:
Abstract
Define BEA zeolite catalyst
Revise all manuscript
Results and Discussion
Define BEA, FAU, MFI MOR and MWW zeolites. Difference between these zeolites. Comment the values of pore diameter (Å) with major detail. Figure 4
Because the separation in two figures. BEA, FAU and MWW, MOR, MFI
[Response] According to reviewer’s comment, the authors supplemented the explanation about difference of the zeolites in introduction section. Pore diameters of the zeolites including other physical properties were described in Table 1. The description supplemented in revised manuscript is as below.
“FAU zeolite consists of 12-oxygen rings, with pores of diameter 7.4 Å in tetrahedral configuration, with large pores called supercage at the intersection. MFI zeolite has an unusual structure in which straight pores with 5.4 x 5.6 Å cross sections and zig-zag pores with 5.1 x 5.5 Å intersect each other. The pores of the MWW zeolite consist of two independent 10-oxygen rings. The two-dimensional 10-oxygen ring is sinusoidal and the other is composed of 10-oxygen ring and 12-oxygen ring. The pores of the BEA zeolite consisted of 12-oxygen rings, and the pore size parallel to the [001] plane is small but the pore size of the [100] plane is large. The pores are curved like sinusoids. MOR zeolite has a straight main hole consisting of 12-oxygen rings and pores consisting of 8-oxygen rings there between.”
Revise the presentation of the Figure 6 It is necessary that the authors comment with major detail Scheme 1. The authors comment the uses of “high acid”, which is the acid used.
Other comment:
Page 9, Line 189
CHANGE 0.5 N FOR 0.5 M
Page 9, Line 193
CHANGE SiO2 FOR SiO2
Page 9, Line 195
“After aging for hours” How many hours?
[Response] The authors corrected all the errors pointed out by the reviewer. In addition, incorrectly marked parts of the text and spelling errors were corrected. The authors appreciate the reviewers' meticulous comments.
The authors complemented the introduction part and strengthened discussion in the revised manuscript. The authors revised also grammatical errors and mistakes of spelling.
The authors appreciate again for the opportunity to respond to reviewer comments. Please contact me if you have any questions.
S.-C. Jung
Sunchon National University, Korea

Round 2
Reviewer 1 Report
The authos have change part of text as required frem the reviwer, for this reason I agree to accept the article in the present form.